# Global field observations of tree die-off reveal hotter-drought fingerprint for Earth's forests

William M. Hammond [1✉], A. Park Williams [2], John T. Abatzoglou [3], Henry D. Adams[4], Tamir Klein [5], Rosana López [6], Cuauhtémoc Sáenz-Romero [7], Henrik Hartmann [8], David D. Breshears[9] & Craig D. Allen [10]

Earth's forests face grave challenges in the Anthropocene, including hotter droughts increasingly associated with widespread forest die-off events. But despite the vital importance of forests to global ecosystem services, their fates in a warming world remain highly uncertain. Lacking is quantitative determination of commonality in climate anomalies associated with pulses of tree mortality—from published, field-documented mortality events—required for understanding the role of extreme climate events in overall global tree die-off patterns. Here we established a geo-referenced global database documenting climate-induced mortality events spanning all tree-supporting biomes and continents, from 154 peer-reviewed studies since 1970. Our analysis quantifies a global "hotter-drought fingerprint" from these tree-mortality sites—effectively a hotter and drier climate signal for tree mortality —across 675 locations encompassing 1,303 plots. Frequency of these observed mortality-year climate conditions strongly increases nonlinearly under projected warming. Our database also provides initial footing for further community-developed, quantitative, ground-based monitoring of global tree mortality.

[1] Agronomy Department, University of Florida, Gainesville, FL 32611, USA. [2] Department of Geography, University of California, Los Angeles, Los Angeles, CA 90095, USA. [3] Management of Complex Systems, University of California, Merced, CA, USA. [4] School of the Environment, Washington State University, Pullman, WA, USA. [5] Department of Plant and Environmental Sciences, Weizmann Institute of Science, Rehovot, Israel. [6] Sistemas y Recursos Naturales, Universidad Politécnica de Madrid, Madrid, Spain. [7] Instituto de Investigaciones sobre los Recursos Naturales, Universidad Michoacana de San Nicolás de Hidalgo, Morelia, Michoacán, Mexico. [8] Department of Biogeochemical Processes, Max Planck Institute for Biogeochemistry, Jena, Germany. [9] School of Natural Resources and the Environment, University of Arizona, Tucson, AZ, USA. [10] Department of Geography and Environmental Studies, University of New Mexico, Albuquerque, NM, USA. ✉email: williamhammond@ufl.edu

Central to global ecosystem services and human economies, forests serve as keystone habitats for life, ecosystem drivers for the cycling of water and carbon, and both structural and economic support for human civilization[1]. Global forests are composed of >60,000 tree species[2], store nearly half of terrestrial carbon, and sequester up to a third of anthropogenic annual carbon emissions[3]. Earth's historical forests (large-tree communities with old-growth dominants established before circa 1880[4]) are disproportionately vital in the cycling of carbon and water, and in supporting biodiversity[5,6]. Anthropogenic change poses many threats to forests—wildfires[7], deforestation, and especially hotter drought (i.e., drought during newly emerging chronic and/or acute hotter climate conditions[8,9]) have impacted some of these giants of ecosystem services on Earth. Understanding which forests will persist—or die-off—is urgent. Quantification of common climate drivers of tree mortality, across biome-specific differences among Earth's forests, may help identify future drought mortality risks that are scalable globally. Despite the widespread association between hotter drought and tree mortality events (events when mortality was significantly increased from expected background rates[10,11]), it has not been possible to quantify the climate conditions triggering forest die-off globally without a more precise record of where and when tree mortality events have occurred on Earth. Linking global field observations of tree die-off to climate is urgently needed to validate model projections that collectively suggest the substantial future vulnerability of forests to die-off[11–13]—but current extrapolations are based on: experimental manipulations such as rainout shelters, whole tree chambers, or greenhouse experiments[14–18], single site or regional-scale climate conditions[8,13,19,20], and process-based projections ranging from site-specific physiological models to global Earth system models[21] that still need better-parameterized and more-realistic representation of tree mortality processes[9,11,12,22]. Thus, forests globally—especially Earth's historical forests, relics of recruitment under already-bygone climate conditions—face uncertain fates in today's rapidly warming world.

Rising temperatures present a triple threat to tree survival: amplification of atmospheric drought, intensified soil drought, and direct effects of heat stress. As temperatures rise, so too does the vapor pressure deficit (VPD, a measure of atmospheric drought), accelerating water loss from both soils and trees during hotter periods, even when leaf stomata are closed[23,24]. Anthropogenic warming also is increasing the frequency, severity, and intensity of chronic soil droughts[25]—and diverse evidence from tree rings to remote sensing[13] documents both antecedent warning signals of tree mortality[26], and lagged tree growth and mortality effects, with chronic droughts[27]. Direct effects of soil drought on trees can be observed in their physiological responses—as water becomes scarce, global observations show that trees limit water loss via stomatal control[28], and deploy diverse (e.g., stomatal, osmotic) adjustments to ameliorate the immediate effects of drought[9]. Mild drought merely reduces growth and impairs physiological functioning, but severe drought can permanently damage plant physiological function and even become lethal when basic hydration and/or metabolic needs are not met—resulting in plant tissue collapse and eventually death[16,29]. Heat stress can also directly impair plant function and metabolism, and at sufficient intensities is even lethal[15]. Although plants have some ability to acclimate to warming during drought stress[15], this acclimation potential can be overwhelmed by either sufficient chronic or acute warming and/or by a drought of sufficient severity[8,9]. Hotter droughts present a deadly trade-off to trees, where using water to ameliorate heat stress via evaporative cooling must draw from the same shrinking pool needed to survive the drought[9,29]. Thus, intensifying atmospheric drought due to

warming is especially water-costly to trees concomitantly experiencing long-term deficits in soil moisture.

Unfortunately for many of Earth's forests, the frequency of co-occurring extremes of heat and drought has increased over the past century primarily due to warming climate[30,31]—the highest-confidence and most pervasive global signal of anthropogenic climate change. Climate variability leads to droughts, including hotter droughts, but climate change fundamentally alters the character of these droughts[25]. At some point, increasingly extreme hotter droughts exceed the historical climatic limits to which forests are locally adapted, overwhelming forest resilience and resulting in amplified mortality across diverse biomes. We hypothesize that this climate exceedance has begun to emerge in recent decades, and may be detectable with a set of hotter-drought metrics referred to hereafter as a "global hotter-drought fingerprint" of climate change on pulses of tree die-off. Thus the question: can we determine a global fingerprint (a common set of climate anomalies linked to globally-distributed field-observed tree die-off) of hotter-drought-triggered tree mortality, and if so, what climate conditions constitute such a global fingerprint—how hot is too hot, and how dry is too dry, relative to long-term climate? To answer these questions, we aimed to (1) establish a precisely geo-referenced database of on-the-ground climate-induced tree mortality observations; (2) use this database to quantify a global hotter-drought fingerprint on Earth's forests; and (3) determine changes in the frequency of climate conditions recently associated with tree die-off under further warming.

## Results

**Database of global tree mortality.** Here we provide a global database of precisely geo-referenced observations of tree mortality associated with drought and heat (but excluding fire), from 154 peer-reviewed publications (Table S1 and Supplementary Data 2) spanning five decades, including research on all of Earth's forested continents (see "Methods"). Our literature review and data requests resulted in a global database of 1303 plots (Supplementary Data 1) where ground-based field observations of drought and/or heat-induced tree mortality occurred between 1970 and 2018 (Fig. 1). While our database is inherently biased by our use of only the available peer-reviewed studies documenting where forests experienced climate-induced die-off in the field, it nonetheless represents the only globally cohesive and precisely georeferenced plot-level set of peer-reviewed observations for climate-induced tree die-off. While our database sites have a strong northern hemisphere bias (70%, $n = 471$ of the 675 total sites), the total forested area in the northern hemisphere is ~78% of the global total (calculated from Simard et al.'s canopy height shown in Fig. 1). Even so, critical carbon sinks, particularly boreal forests and both tropical and temperate rainforests, remain under-sampled. In the absence of globally coherent and standardized forest monitoring sampling designs, this database presents an opportunity to ask planetary-scale questions about the fates of Earth's forests under changing climate conditions.

Every global biome with trees is represented in our database of climate-driven tree die-off (Fig. 2). Mortality sites spanned an annual precipitation gradient of over 4 m, an annual average temperature gradient of over 30 °C, and elevations from sea level to 3488 m (Fig. 2). By Whittaker biome[32], woodland/shrubland accounted for 49% of plots ($n = 638$) which spans a large climatic niche. We note that these coarse climate-based Whittaker biomes can obscure heterogeneous forest types within single biomes; in particular, the woodland/shrubland Whittaker biome is dominated in our database by diverse, relatively dry but often closed-canopy forest types, including those composed of aspen and

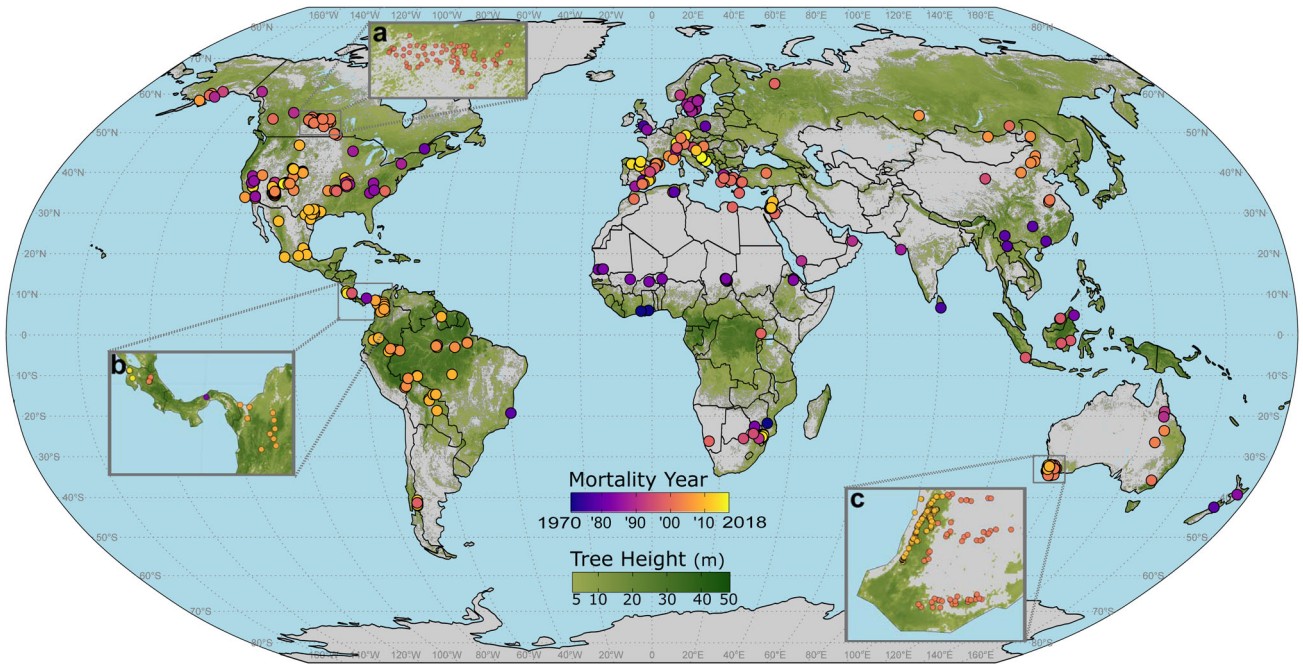

**Fig. 1 Global distribution of hotter-drought tree mortality plots.** Geo-referenced tree mortality plots ($n = 1303$) in the database. Dots are color coded according to the year of mortality. Each point has been precisely georeferenced. Insets show examples of dense plot networks in Canada, Central America, and Southwest Australia. To illustrate the extent of global forests, a background layer in green shows canopy height[54], with darker green shading indicating taller forests. Inset **a** shows a broad plot network from aspen die-off during drought in Canada. Inset **b** shows dense plot networks in Costa Rica, Panama, and Colombia. Inset **c** shows plots in the Jarrah and Wandoo forests of Southwest Australia.

numerous conifer, oak, and eucalypt species. Temperate seasonal forest plots represented 14% of the database, followed by subtropical desert (13%), temperate grassland/desert (12%), and tropical seasonal forest/savanna (8%). Tropical rainforest, temperate rainforest, and boreal forest biomes combined contained just 4% of plots. The remainder (4 plots) fell outside of Whittaker biome space. Multiple biomes experienced tree mortality in 78% of the mortality years covered by our database (Supplementary Fig. S5).

**Quantifying a hotter-drought fingerprint on Earth's forest mortality sites.** We evaluated six climate metrics from TerraClimate[33], a globally gridded climate and hydroclimate product, coincident with the tree-mortality database: monthly average maximum temperature (TMAX), vapor pressure deficit (VPD), climatic water deficit (CWD), soil moisture (SOIL M), monthly total precipitation (PPT), and the Palmer Drought Severity Index (PDSI). Our 1303 plots were encompassed within 675 locations at the spatial resolution (1/24°) of TerraClimate. We chose these six climate variables a priori from the TerraClimate database due to their being direct or indirect measures for heat and/or drought impacts. To quantify a hotter-drought fingerprint (e.g., climate extremes associated with heat- and drought-induced tree mortality), we identified the climatological warmest and driest months (e.g., months with the highest TMAX, VPD, and CWD and lowest SOIL M, PPT, and PDSI—to look for intensified extremes for heat and drought, respectively) for each variable at each location during the 61 years of available climate data (1958–2019). Note, while other climatic factors (e.g., duration or seasonal timing of drought/heat stressors) may also be important, our present analysis focuses on detecting acute climatic anomalies at monthly resolution.

We calculated monthly anomalies (standardized as z-scores) of these climate metrics relative to the period of record (1958–2019) to facilitate comparison across the diverse climates at database

sites during the years bounding mortality events (±4 years from the onset of mortality). While individual variables may exceed these mortality year anomalies (e.g., variable measures which are hotter or drier than long-term average during the year of known local tree die-off events) at high frequency (Supplementary Fig. S2), only rarely do all six metrics, which together comprise our hotter-drought fingerprint, exceed their mortality year anomalies.

Across the global database, climate conditions in the year of tree mortality (defined as the year mortality began, as determined from source papers or data requests) for every metric we assessed were significantly warmer and/or drier than the long-term mean (Fig. 3). Specifically, during the mortality year, we identified a global hotter-drought fingerprint when typically warmest/driest months for TMAX, VPD, and CWD z-scores were significantly higher than the long-term average by $0.37\sigma \pm 0.04$ SE, $0.30\sigma \pm 0.04$, and $0.49\sigma \pm 0.04$, respectively—while PPT, and SOIL M z-scores were below the long-term average by $-0.21\sigma \pm 0.03$, and $-0.39\sigma \pm 0.03$, respectively. z-Score magnitude for PDSI was ($-0.73\sigma \pm 0.04$), which includes the memory of water balance anomalies over several months (similar large anomalies were found using 3-, 6-, 12-, and 24-month Standardized Precipitation-Evapotranspiration Index (SPEI) see Supplementary Fig. S1). The year preceding the onset of mortality was also significantly warmer and drier for all variables, although less so than the year mortality began; the year following the onset of mortality also tended toward hotter and drier than the long-term average. In contrast, years preceding or following this 3-year window—centered on the year mortality began—had smaller differences from the long-term average (Fig. 3), suggesting that episodic hot droughts rather than solely long-term trends are responsible for the hotter-drought fingerprint signal.

This hotter-drought fingerprint was also detected across all biomes individually in a separate biome-specific analysis—except for tropical rainforest (see Fig. 4), where our sample of known

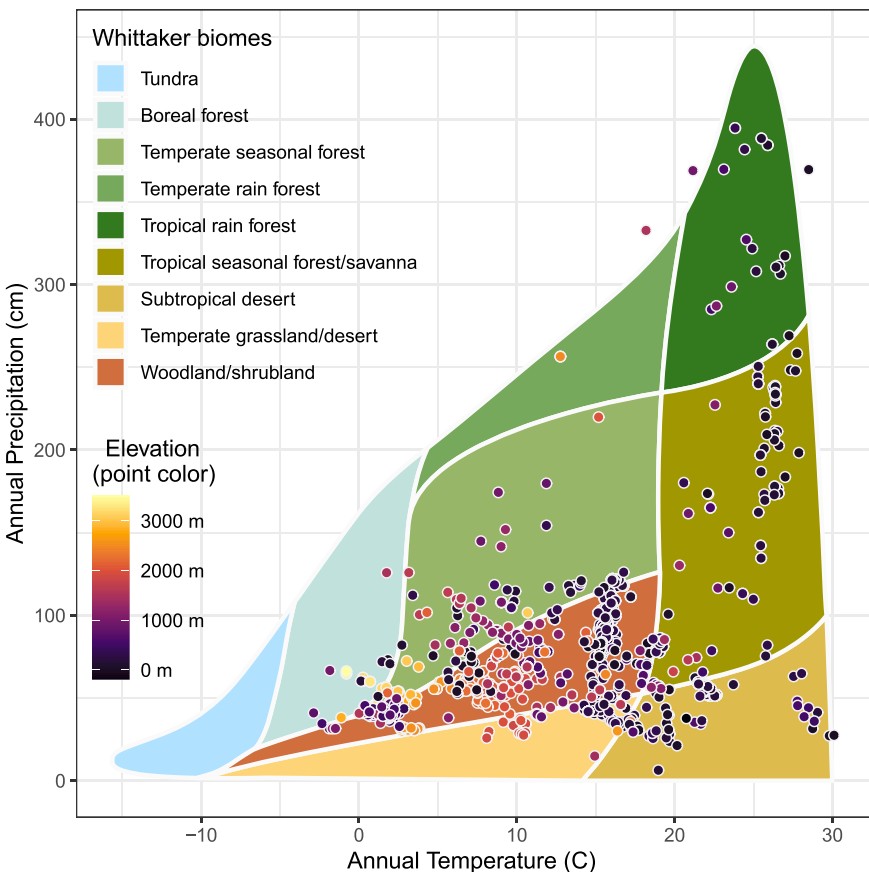

**Fig. 2 Biomes of global tree mortality plots.** Whittaker biome plot of database locations, showing the mean annual temperature (°C) and mean annual precipitation (cm) for all 1303 database plots. Each point represents a plot from the global database where climate-induced tree mortality happened. These mortality locations have occurred in all forested biomes, across a range of 30 °C of mean annual temperatures, and a four-meter annual precipitation gradient. Point color represents elevation, which ranged from sea level to 3488 m. Climate data for mean annual precipitation and mean annual temperature are taken from TerraClimate[33] and represent the average over 1970–2000. These data illustrate that whether a site is typically cool, warm, dry, or wet—eventually, a locally extreme hot drought can lead to tree mortality.

mortality events is limited and sparse climatological instrumental records may result in higher uncertainty for the spatially-interpolated climate data used in our analysis. At the site level, conditions in the mortality year were anomalously hot/dry for all six variables at 24% of sites, for at least 5 variables at 47% of sites, and for at least 4 variables at 69% of sites (Supplementary Fig. S2). While individual variables may exceed these mortality year anomalies at high frequency (Supplementary Fig. S3), only rarely do all six metrics, which together comprise our hotter-drought fingerprint, exceed their mortality year anomalies.

Since 1970, many conditions during mortality years became warmer and drier in the study plots. In particular, TMAX, VPD, and CWD all increased while water balance metrics of SOIL M and PDSI decreased (Supplementary Fig. S7). No change was seen in dry-month precipitation anomalies—which may be partly due to the large proportion of sites that receive zero precipitation during their typically driest month. At our sites, TMAX has been increasing during mortality years faster than background warming (Fig. 5), consistent with continental-scale observations of intensifying extremes[31].

**Further warming increases the frequency of hotter-drought fingerprint climate conditions**. Under the two climate scenarios (+2 °C, +4 °C warming of global mean temperature relative to pre-industrial), superposed on observed climate (1985–2015, global mean temperature +0.7 °C above pre-industrial) for comparison[34], extreme climate conditions are projected to become drier (more negative PDSI), hotter (higher TMAX anomalies), and more arid (higher VPD) relative to observed climate (Fig. 6a–c). We determined the number of years when each of the six variables exceeded their site-specific mortality-year heat and aridity thresholds—the local, site-specific hotter-drought fingerprint (these are years when all six variables were each exceeded). We found that the frequency of mortality-triggering extreme climate conditions increases nonlinearly with warming (Fig. 6d). Under the observed (1985–2015) climate, mortality-year hotter-drought fingerprint climate conditions occurred on average 1.62 years per decade (+/−0.08 SE) at sites in our analysis. Under +2 °C and +4 °C scenarios, mortality-year climate condition frequencies increase by 22 and 140% (1.97 +/− 0.07, 3.88 +/− 0.10 years per decade), respectively.

## Discussion

**Earth's forests imperiled by further warming**. We quantified a global-scale hotter-drought fingerprint, representing a global climate signal for years with documented site-specific tree mortality. Climate-induced tree mortality in recent decades under hotter-drought conditions has been documented across forests from a diverse array of boundary conditions, spanning from the tropics to the boreal, from sea level to 3,500 m, and across a four-meter precipitation gradient and 30 °C of mean annual temperature. One reason that the hotter-drought fingerprint is similarly evident in the year prior to reported mortality onset (Fig. 3), as well as largely echoed in the year after, may be due to the imprecise nature of

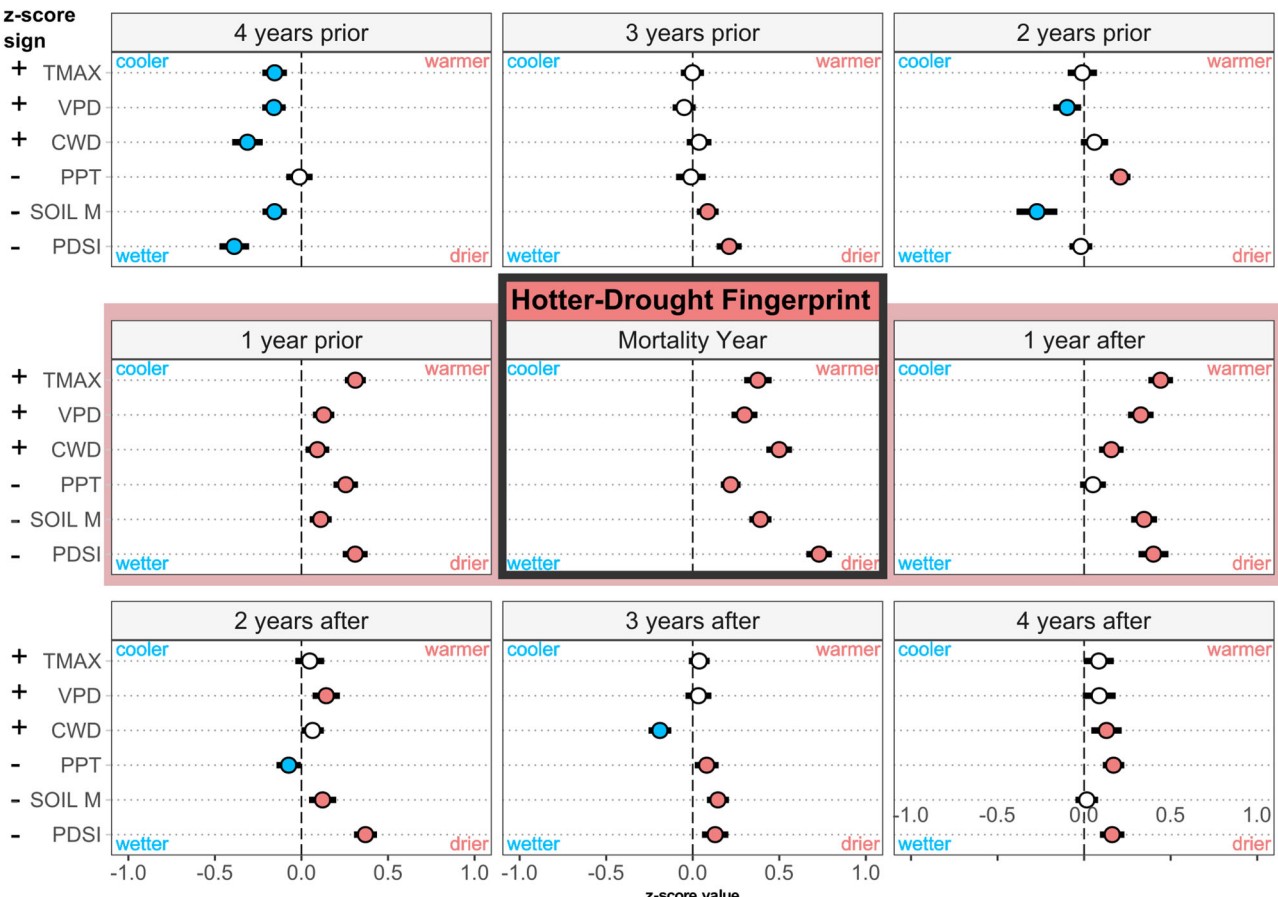

**Fig. 3 Hotter-drought fingerprint of global tree mortality.** The *z*-scores of climate variables during climatological warmest/driest months for the 4 years preceding, 4 years following, and during the onset of tree mortality events. For each variable, circles indicate the mean *z*-score of all sites (*n* = 675), while whiskers represent the 95% confidence interval of that mean. Panels are arranged chronologically from left to right, top to bottom, from 4 years before mortality began through 4 years after. During the year of onset of mortality, a hotter-drought fingerprint on forest mortality events was revealed as the mean of plot *z*-scores for all climate variables significantly shifted toward values characteristic of warmer and drier conditions, with the year prior and the year following mortality showing similar, but weaker, tendencies (this 3-year window, centered on the year mortality began, is outlined in red). The *z*-scores for TMAX, VPD, and CWD are shown with their original sign, while the sign of *z*-scores for PPT, SOIL M, and PDSI were flipped such that positive indicates warm/dry, and negative indicates cool/wet across all variables. Point color represents the variable condition relative to long-term (1958–2019) climate: white indicates no difference, blue is significantly wetter/cooler, and red is significantly warmer/drier.

identifying the "onset" and duration of mortality (e.g., visual indications of mortality may lag significantly behind environmental drivers[16]). In addition, chronic drought conditions commonly span multiple years, cumulatively predisposing eventual, lagged mortality events[13,26,27]—consistent with our observed "3-year hotter-drier window," centered on the nominal mortality year (Fig. 3).

Our global-scale hotter-drought fingerprint, focused on acute hotter-drought extremes, represents a cohesive signal for climatic drivers of tree die-off in many of Earth's forests. Other approaches could consider other temporal dimensions of climate signals (e.g., shorter-term heat-wave stress, longer-term chronic drought, changes in seasonal drought duration or timing), which may further improve our understanding of climatic drivers of tree mortality. Ideally, future efforts to harmonize global forest inventory and monitoring methodologies, including their currently-disparate documentation of tree mortality, will reduce the inherent sampling biases (typically favoring northern hemisphere and/or areas adjacent to well-funded research institutions) and presence-only limitations of our present database[11].

Additionally, we found that many of Earth's forests may become increasingly imperiled by further warming and drought, as the frequency of lethal climate conditions observed with recently

documented global mortality events will accelerate with further warming (Fig. 6d). Although our approach does not reveal the particular detailed mechanistic ecophysiological responses to the hotter drought that are driving mortality for each specific site, it exemplifies the powerful utility and practical potential of empirical approaches that link direct observations of tree mortality from diverse precisely georeferenced locations to observed climate drivers. While multiple emerging lines of evidence indicate that warming puts trees at greater risk under drought conditions[9,14,15,19,24,35], the quantitative hotter-drought fingerprint we identified here suggests that further warming may accelerate global forest die-off across many biomes. The impact of this hotter-drought fingerprint is acting on Earth's forests already, with nearly half a billion trees having died from hotter-drought events in Texas and California alone since 2010[36,37]. In central Europe, hotter drought starting in 2018 has led to extensive dieback of forests that is ongoing—and of yet undetermined magnitude and extent—which could lead to significant ecological transitions[38]. Other notable global tree mortality events documented during hotter-drought episodes include three pulses of large-tree mortality since 2005 across Amazon basin tropical moist forests[39,40], and historically unprecedented hotter-drought-triggered dieback in Jarrah forests of southwest Australia in 2011[8,19].

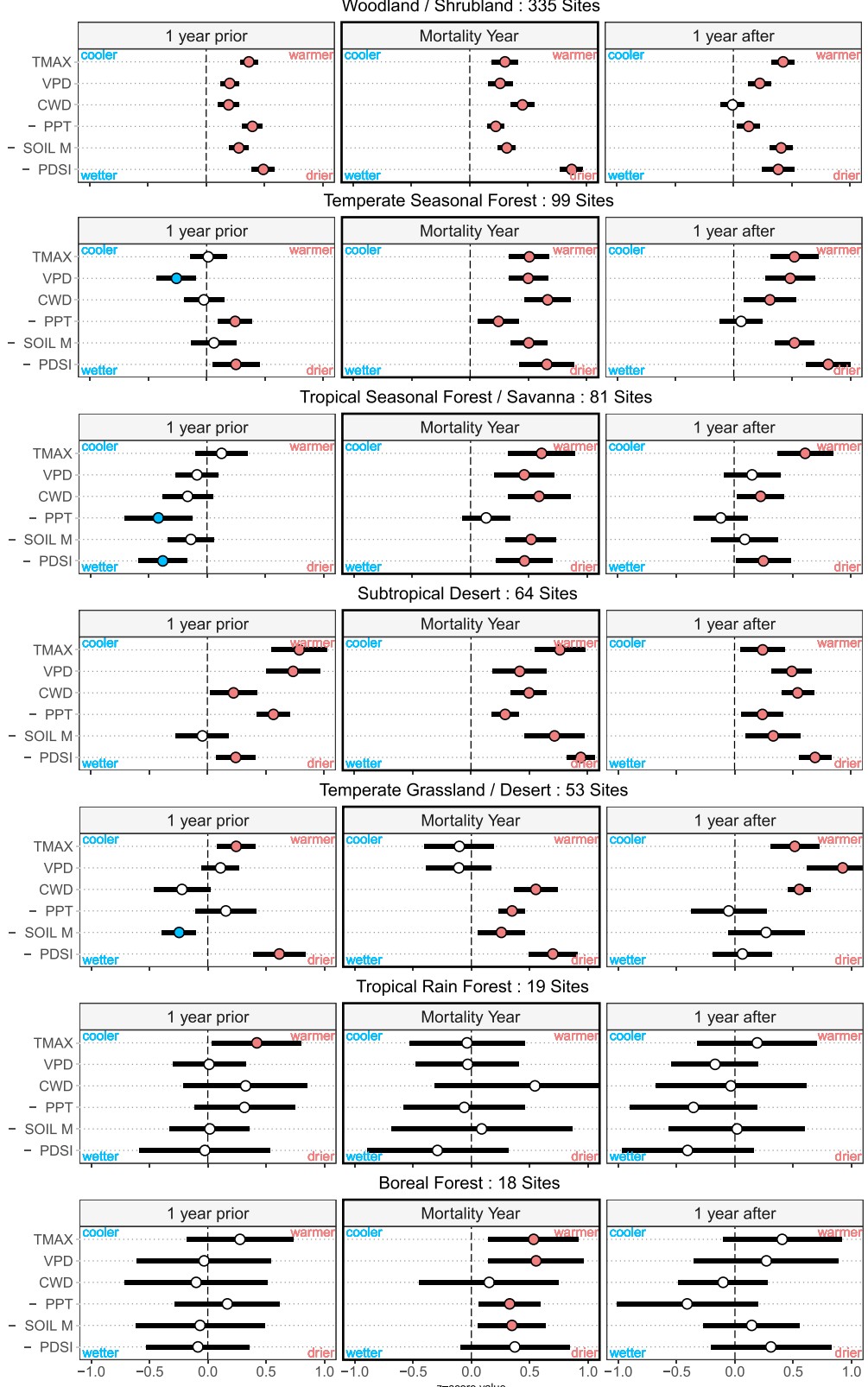

Individual trees and forest ecosystems may benefit in various ways (e.g., increased water-use efficiency, stored non-structural carbon, etc.) from productivity gains under elevated atmospheric $CO_2$[22]—when soil nutrients and water are not limiting. However, the net effects of increasing $CO_2$ in combination with a changing climate on the mortality of global forests during hotter drought are uncertain[4,9,35]. In particular, during hotter-drought events, plant uptake of $CO_2$ is limited by the initial closing of stomata—with $CO_2$ uptake eventually blocked as leaves lose turgor, followed by failure of the coupled plant water-and-carbon transport system which may ultimately result in death[16,28]. Thus, potential amelioration of tree mortality risk by the ~85 ppm

**Fig. 4 Hotter-drought fingerprint by Whittaker biome type.** The 3-year window, centered on the mortality start year, is displayed for each biome. Above each biome triptych the number of sites included is listed (Woodland/Shrubland $n = 355$, Temperate Seasonal Forest $n = 99$, Tropical Seasonal Forest/ Savanna $n = 81$, Subtropical Desert $n = 64$, Temperate Grassland $n = 53$, Tropical Rain Forest $n = 19$, and Boreal Forest $n = 18$). Points represent the mean z-score for each variable, and whiskers are the 95% confidence interval in that mean. Strong signals appear in woodland/shrubland, temperate grassland/ desert, both temperate seasonal forests, and tropical seasonal forests/savannas, and even boreal forests (with a small sample) show significant indicators of hotter drought. Tropical rainforests do not show this particular hotter-drought signal in our present analyses—highlighting that a combination of data scarcity and diverse responses of various forest types in the tropics will need further investigation, as outlined in point (5) of the future research opportunities listed in the main text.

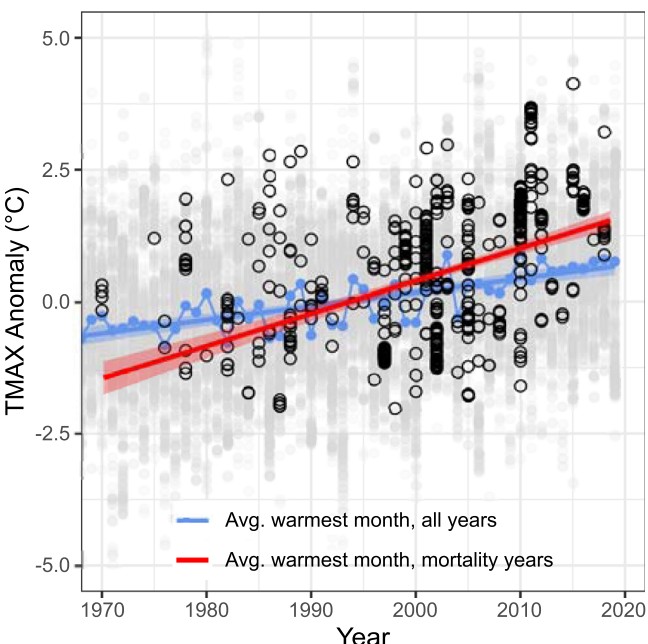

**Fig. 5 Mortality year TMAX warming faster than all years.** Linear regressions showing the trend of TMAX anomalies during the typically warmest month across all years (1970–2018, blue trend line is model fit of the linear regression) and mortality event years in the database (red trend line is model fit of the linear regression, as in Supplementary Fig. S7A). Blue dots show the mean values for TMAX anomaly during the typically warmest month, for all years at all sites, and the faded gray dots the raw data. Black open circles indicate mortality year TMAX anomalies for each site. For both regressions, gray shading represents the standard error of model fit. TMAX Anomalies are increasing faster during mortality event years (red line), than during the chronic background warming illustrated by the all-years, all-sites regression (blue line).

atmospheric $CO_2$ increase during the timeframe in our database (1970–2018) might have been overwhelmed by the concurrent increases in temperature during mortality-event years (Fig. 5). This warming presents a triple threat to tree survival in the form of amplified soil drought, atmospheric drought, and heat stress, and our results are consistent with experimental findings that drought and warming can negate or overcome the effects of elevated $CO_2$[17,18].

**Earth's historical forests are especially vulnerable**. As the longest-lived organisms on Earth, trees routinely are imbued with historical and cultural significance by human societies, while also persistently sequestering carbon and amplifying local biodiversity for centuries, sometimes millennia. In contrast, extreme climate stress events occur on the scale of days to months to a few years, and in these relatively brief periods, large old trees—exemplars of Earth's historical forests[6]—can be especially susceptible to mortality[5,41–44]. Forests will certainly persist and thrive over large

areas into Earth's future, but increasingly they will have to rapidly shift in physiological function, morphology, genetics, species composition, structure, and geographic distribution in response to anticipated climate changes. Where the pace of climate change outruns the adaptive or acclimation capacities of historically-dominant tree individuals and species, additional die-off events are likely to occur and some forests may even cease to exist. In particular, the current tree communities of Earth's historical old-growth forests—which took centuries, sometimes millennia, to grow to structural dominance under now locally-shifted climate conditions—may continue to often be most negatively affected by continued warming and drying[4,43], as novel hotter-drought extremes increasingly exceed their range of survivable climate across diverse forested biomes. The expected near-term outcome is simplified tree communities, where more drought- and heat-tolerant species survive, and less tolerant species diminish or perish. In many cases, this may lead to lasting changes in vegetation composition, stature, and spacing, where surviving woody plants in these communities do not maintain or develop the complex canopy structure typical of historical old-growth forests[4,9,35,45].

**Underestimation of tree mortality from hotter droughts**. While our projections for an increase by up to 140% in the frequency of climate conditions associated with recent forest die-off under $+4\,°C$ may seem severe, they are modest in comparison to some current empirical and mechanistic process-based model predictions for catastrophic forest die-off at continental scales under hotter droughts[12,14]. Our projections for increasing die-offs under further warming are consistent with projections showing the potential for large increases in mortality under future hotter drought[12,14,46], although these projections are often limited to single species or single biomes. Even continental-scale projections for up to 40% increases in the frequency of mortality-inducing hotter droughts under $\sim+2.5\,°C$ since pre-industrial[20] are in general agreement with our global analysis's 20% under $+2\,°C$ (Fig. 6d). Further, our projections of increasingly frequent, historically lethal climate conditions for Earth's forests may be conservative for several reasons:

(1) Requiring that all six climate variables meet or exceed mortality year conditions, concurrently in the same year, is a strong filter. For example, TMAX, VPD, and PDSI all exceed mortality-year conditions under $+4\,°C$ in about 4 out of every 5 years (Supplementary Fig. S3), whereas under the same warming scenario, all six metrics exceeded the hotter-drought fingerprint only half as often.

(2) Tree mortality involves diverse disturbance processes that amplify forest die-off in the presence of global warming and hotter droughts[4,24,35] but these were excluded in our analysis, including insects[44,47], pathogens[48], wind[40,49], and lightning[50]. Additionally, anthropogenic warming promotes greater wild-fire activity, particularly fire extent and severity in many forests worldwide[7,51], driving further declines in some of Earth's forests. We also have not considered disturbance

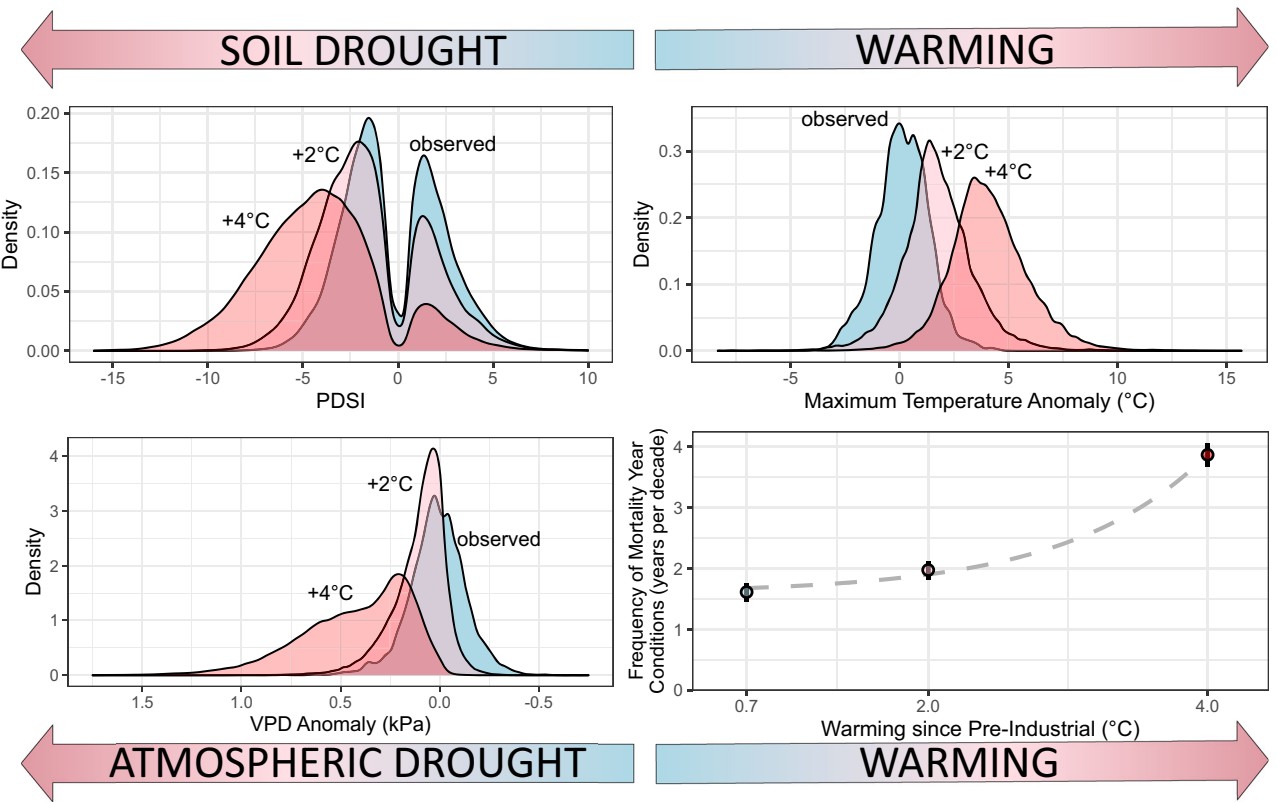

**Fig. 6 Warming triple-threat accelerates mortality risks.** Under two warming scenarios, leading indicators of tree mortality representing the triple-threat that warming poses to tree survival at the plots in our analysis departed from recent values toward drier (PDSI (**a**), VPD (**c**)) and warmer (TMAX values, **b**) indicating further warming will shift climatic space toward that recently associated with mortality events. In **a**–**c**, the density distributions of PDSI, TMAX, and VPD during observed (+0.7 °C, light blue) and warmed +2 °C (pink) and +4 °C (red) climate scenarios are shown. In **d**, open circles show the mean ($n = 675$ sites) frequency of mortality year hotter-drought fingerprint conditions (when for a particular location, all 6 monthly variables met or exceeded the local hotter-drought fingerprint conditions of the mortality year), while the x-axis represents the mean warming since pre-industrial (1850–1879) climate. Error bars represent 95% confidence interval and the dashed gray line is an exponential fit.

interactions among these many amplifying and synergistic agents of tree mortality[49,52]—but conversely, we also acknowledge that thinning from either climate-triggered mortality or these associated synergistic agents, may partially buffer against future losses[35,45].

(3) Our findings indicate that climate anomalies of tree mortality event years are trending towards ever hotter and drier conditions (Fig. 5, Supplementary Fig. S7), concurrent with any potential ongoing forest acclimation to temperature and/or $CO_2$ fertilization[15,22]. Yet the potential for tree species to acclimate to ongoing climate warming, even with increasing atmospheric $CO_2$ concentrations, is not unlimited—and when exhausted—forest die-off may rapidly accelerate[9,35,53]. Since projected warmer climate conditions include unprecedented extremes of hotter drought for which there are no observed analogs, the potential for crossing historically unknown tipping-point climatic stress thresholds may increase, further amplifying tree mortality[35].

(4) Our analysis of mortality-year frequency uses monthly climate data, yet important drivers can occur on longer (e.g., drought[26]), and shorter (e.g., heatwave[8,19]) timescales. For example, the 4-year-prior signal of cooler/wetter climate (Fig. 3) may reflect favorable pre-drought conditions promoting structural overshoot of trees, which could amplify dieback and mortality risk during subsequent years of hotter drought[45].

**Roadmap for research enabled by a quantitative ground-based global database.** The widespread global coherence of our empirically quantified hotter-drought fingerprint may provide immediate opportunities to validate projections of tree mortality in existing models of the Earth system, while also enabling diverse future analyses. Although global in geographic extent, our database is limited by the availability of peer-reviewed, ground-based empirical studies of climate-induced tree mortality, and thus only sparsely covers some regions, particularly large portions of boreal and tropical forests. For example, our hotter-drought fingerprint was consistent across all biomes except the tropical rainforest (Fig. 4)—despite published direct observations of hotter drought as a driver of tree mortality at these tropical rainforest sites[39,40]. Additionally, this biome may experience pulses of tree mortality in response to different climate fingerprints, particularly involving longer-duration dry seasons—not just intensified single monthly extremes.

Despite this and some other limitations, our database represents a globally-distributed dataset with precisely geo-referenced sites where ground-based heat- and drought-induced tree mortality has been documented. Our use of this database to quantify a global hotter-drought fingerprint of tree mortality illustrates the potential for rapid progress in empirical modeling of forest mortality drivers and thresholds at spatial scales from local to global, where direct observations of forest responses to climate stress can help identify and quantify mortality drivers. Toward the goal of fostering further rapid community development of many more such direct observational records of climate-

induced forest stress and tree mortality worldwide—with methods ranging from local ground-based sites to synoptic remote-sensing—this database immediately will be served as an open-access resource at the International Tree Mortality Network (https://www.tree-mortality.net), an academic networking initiative associated with the International Union of Forest Research Organizations' (IUFRO) task force on monitoring global tree mortality patterns and trends (https://www.iufro.org/science/task-forces/tree-mortality-patterns). The complete database—along with an interactive version of Fig. 1 from this paper—will allow users to zoom in on dense plot networks, with direct links to the supporting literature for each point. This online database includes the reference for each plot, its precise coordinates, dominant species, associated biotic agents, and the year of mortality onset. To further update and rapidly increase the quantity and spatial representativeness of global tree mortality observations, ongoing online contributions from diverse observer groups, ranging from practicing foresters and field ecologists to remote-sensing scientists, can be integrated into the website in near-real-time via a user-friendly entry form.

As the only global set of ground-truthed observations of drought- and heat-induced tree mortality, this database can immediately aid in validating remote-sensing technologies for eventual synoptic monitoring in near-real-time of tree mortality (which could then feedback into the database). Additional groups to benefit from the database are those interested in climate and physiological mechanisms of tree mortality, including the connected fates of all forest-dependent life[5,19], with an aim toward improving the representation of climate-induced tree mortality representations in Earth system models. Related future research opportunities associated with this initial online database include:

(1) Identify additional chronic (e.g., seasonal to decadal) and acute (daily to weekly) climatic signals of tree mortality, including thorough analyses that quantitatively consider antecedent and lagging factors, and duration and seasonality of drought stress;

(2) Synthesize mortality observations from extensive forestry plot inventory networks, to increase spatial representation for the global climate signal of tree mortality, and to identify where during these events trees did not die-off;

(3) Apply remote-sensing approaches to mortality detection using this spatially precise (and in some places plot-dense) database for ground-truthing, to determine the full spatial extent of known mortality events, and aid in ongoing monitoring of forest stress and tree mortality events in near-real-time;

(4) Benchmark state-of-the-art Earth system models via hindcasting, to assess the accuracy of tree mortality event representation—and to do so across spatial resolutions (as in Supplementary Fig. S4) at which these planetary models operate;

(5) Develop approaches to understand potentially unique features and drivers of hotter-drought mortality in tropical rainforests (differing climate signals, e.g., extended dry seasons, where warming/drying of typically moderate shoulder seasons may matter more than intensified single-month extremes), the single biome in which our global approach did not reveal a strong hotter-drought fingerprint;

(6) Investigate how the severity of forest die-off events will respond to further warming; and

(7) Invest in monitoring, documenting, and gathering mortality data for forests under-represented in this initial global database—especially in the extensive critical carbon sinks of boreal forests and tropical rainforests.

**Future challenges for Earth's forests and societies under hotter drought.** In conclusion, our findings reveal the emergence of a global acceleration of lethal climate conditions, associated with recent forest mortality events, under further warming. Earth's historical forests in particular face a challenging future, including dramatic changes in the extent, composition, age, and structure of these unique and irreplaceable forests, with planetary-scale consequences for biodiversity and the cycling of water and carbon. Our findings both corroborate earlier studies of hotter-drought driven mortality at local to regional scales[8,13,19,20,24,36,38] and extend these findings by quantifying the commonality in climate anomalies across this planetary-scale observation-based database of tree die-off. Although forests often are invoked as an important part of the solution to the present global climate crisis, their role as reliable carbon sinks in mitigating climate change depends upon their ability to survive further warming[10,22,52]— which our global hotter-drought fingerprint identifies as an imminent threat. Our findings show that limiting warming to +2 °C over pre-industrial levels could reduce the frequency of these climate conditions associated with observed tree mortality events to less than half that predicted at +4 °C. Efforts to protect the world's climate from excessive warming likely will be decisive in determining the future persistence of many of Earth's forests.

## Methods

**Literature review methods.** We reviewed the references from four progressively updated recent reviews of drought and heat-induced tree mortality[1,10,11,35] which included references to 209 peer-reviewed studies documenting drought and heat-induced tree mortality. Additionally, we reviewed 21 recent peer-reviewed studies not included in those prior reviews. Studies were included in the database when they met the following conditions: (1) The study had on-the-ground observations of pulses of tree mortality (e.g., events where mortality was significantly increased from expected background rates). (2) The study attributed the mortality to a climatic driver of heat, drought, or their combination. (3) The study contained either precise coordinates (within a kilometer) or a site description, map, or other means of accurate geolocation. To identify the year that significant tree mortality began for a site, we used the authors' assessment (as described in their paper, or communicated during data request). Of the 230 papers, 154 met our criteria and were included in the global database (Table S1).

**Precise georeferencing.** Georeferencing was done in one of two ways: either a paper contained precise coordinates for the location of plots with dead trees, or we submitted a data request to the paper's author seeking coordinates, plot descriptions, and accompanying mortality data. Of the 1303 plots included in our database, 248 had precise location listed in the publication (or associated supplementary materials) while 1055 locations were obtained via e-mailed data requests to authors of the studies. When mortality was reported as a percentage, we determined "higher-than-expected" mortality based on the authors' expertise of their study system. Often, "standing dead" of ≥15% (for mature trees) was used, as 2–3% annual background mortality rates accumulated for up to 5 years would not be expected to exceed this threshold. Of the 76 excluded studies, most were remote-sensing studies (e.g., aerial photographs, aerial observer mapping, satellite or airplane-borne multispectral sensors) or extensive forestry plot inventory networks (e.g., USFS FIA, EU NFIs), where precise geolocation and attribution of mortality to drought and/or heat were not possible. To place these broadly distributed mortality sites in the context of global forests, we plotted our database (in Fig. 1) along with a global canopy cover of ≥5 m[54].

**Historical climate data.** Climate data for our study came from TerraClimate, a globally gridded (1/24-degree, or ~4 km) product of climate and hydroclimate[33]. As the spatial resolution of the climate data was coarser than the spatial resolution of our database, we filtered our 1303 plots with a spatial grid of TerraClimate's resolution, such that 675 unique locations were included in the final analysis, to avoid over-representation of dense plot networks in our analysis. We considered 6 monthly TerraClimate variables from 1958 through 2019: TMAX, mean daily VPD, CWD, SOIL M, PPT, and the PDSI. We additionally considered the 12-month SPEI, calculated from reference evapotranspiration (PET) calculated using the Penman–Monteith approach and PPT[55]. Additionally, we used mean annual precipitation and mean annual temperature data from TerraClimate, from the period 1970–2000, to calculate the Whittaker biome classification[32] of all 1303 plots in our database using R package "plotbiomes"[56]. We acquired elevation data using the R package "elevatr"[57].

**Climate warming data**. We also assessed projected climate for two pseudo-global warming scenarios that perturb TerraClimate data for 1985–2015 with projections commensurate with global mean temperature +2 °C and +4 °C above pre-industrial (1850–1879) levels[34]. These levels of warming are consistent with estimates for "middle of the road" scenarios (e.g., SSP2-4.5) which could result in >+2 °C warming by 2100, and "fossil-fueled development" scenarios (SSP5-8.5) which could result in >+4 °C warming by 2100[58]. Briefly, these warming scenarios use a pattern-scaling approach that scales local changes and variance in monthly climate relative to changes in global mean temperature. These changes are then superposed to 1985–2015 data to provide time-series data comparable with the observational record; we additionally apply an empirical correction to PET calculations that account for enhanced water-use efficiency and enhanced surface resistance to transpiration losses[59].

**Statistical methods for historical data**. For each climate variable (TMAX, VPD, CWD, SOIL M, PPT, PDSI), we calculated the anomaly (for example, TMAX of the warmest month in the year of mortality - TMAX average for the 61 values of the same month from 1958–2019). To provide cross-comparison between metrics with different scales and across disparate climates, anomalies were standardized into z-scores, such that time series had a mean of 0 and a standard deviation of 1 based on using the entire period of record 1958–2019.

We quantified the global signal of mortality by calculating the mean and 95% confidence interval of z-scores for all 675 sites during the year of mortality. To assess the antecedent and lagging conditions for this climate signal, we calculated anomalies and z-scores for the 4 years prior and 4 years following mortality for each plot, and calculated mean and 95% CIs as during the mortality year. We expected this nine-year period, centered on the mortality year (the response for which provides the hotter-drought fingerprint), to include conditions sufficiently prior to mortality (4 years before) to represent normal conditions. As we could only identify the initial year of mortality (studies rarely persist for the years required to determine whether elevated mortality has ceased), we investigated the lagging years to see whether the climatic signal of tree mortality persisted beyond the mortality year, as legacy effects have been reported following intense droughts[27]. We conducted linear regressions for the year of mortality and each variable's anomaly to determine whether anomalies during extreme conditions the year of mortality were increasing, decreasing, or remained unchanged through time.

**Statistical methods for warming scenarios**. We first calculated for each site the number of years during observed (1985–2015) climate in which the local mortality-year threshold conditions (that is, years when all six climate variables met or exceeded the mortality year conditions during that site's typically warmest/driest months). We repeated this with +2 °C and +4 °C warming scenarios. We summarized the frequency of annually concurrent threshold conditions per decade for all sites under historic, +2 °C, and +4 °C climate conditions (Fig. 6d).

**Reporting summary**. Further information on research design is available in the Nature Research Reporting Summary linked to this article.

## Data availability

The location data for this study are available as the initial dataset in the International Tree Mortality Network's Global Tree Mortality Database (http://tree-mortality.net/globaltreemortalitydatabase) and as Supplementary Data 1. The processed climate data for Figs. 3, 4, and 6 of the main text are available at figshare (https://figshare.com/account/home#/projects/131939). Supplementary Data files 1 and 2 contain the location data for all plots and study-level information, respectively, and are provided with the manuscript and further archived on the figshare link above. Climate data used in this study are from TerraClimate (https://www.climatologylab.org/terraclimate.html) and a custom R script for downloading the data using locations provided in Supplementary Data 1 as input is included at GitHub: https://github.com/wmhammond/GlobalTreeMortality.

## Code availability

Code written for this study is available on GitHub at https://github.com/wmhammond/GlobalTreeMortality.

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

## Acknowledgements

We wish to thank Amanda Schwantes, Ted Hogg, Niels Brouwers, George Matusick, Kenneth Feeley, Nate Stephenson, Adrian Das, Joseph Ganey, Maxime Cailleret, Timothy Assal, Patrick Gonzalez, Ted Feldpausch, Erika Gómez-Pineda, and Oscar Trejo-Ramírez for substantial database contributions and assistance in identifying plots suitable for addition to the database. Additionally, we extend our gratitude to the RAINFOR and ForestPlots.net team for their contributions. We also acknowledge Collin Haffey and Alexandra Lalor for early contributions to a smaller, non-georeferenced database of global tree mortality observations. W.M.H. was supported by NSF GRFP #1-653428. H.D.A. was supported by the NSF Division of Integrative Organismal Systems, Integrative Ecological Physiology Program (IOS-1755345), and USDA National Institute of Food and Agriculture (NIFA), McIntire Stennis project WNP00009. C.S.-R. was supported by the Coordinación de la Investigación Científica, Universidad Michoacana de San Nicolás de Hidalgo, and the Monarch Butterfly Fund (Madison, WI, USA). D.D.B. was supported by NSF (DEB-1550756, DEB-1824796, DEB-1925837), USGS SW Climate Adaptation Science Center (G18AC00320) USDA NIFA McIntire Stennis ARZT-1390130-M12-222, and a Murdoch University Distinguished Visiting Scholar award. C.D.A. received support from the U.S. Geological Survey's Ecosystems Mission Area and the USGS Climate Research & Development Program. A.P.W. was supported by DOE DESC0022302.

## Author contributions

W.M.H. and C.D.A. designed and collected the data for the georeferenced database. W.M.H. conceived and conducted the climate analysis, with C.D.A., A.P.W., H.D.A., and J.T.A. providing substantial input. C.D.A., R.L.R., C.S.-R., H.H., and T.K. all contributed significant data directly to the database. W.M.H., D.D.B., and C.D.A. wrote the manuscript, with significant input and contribution from all co-authors.

## Competing interests

The authors declare no competing interests.
