## [Peer Review File · Nature Communications]

Reviewers' Comments:

Reviewer #1:

Remarks to the Author:

The aim of this study and manuscript was to describe a database of tree mortality studies in published literature, to assess if there was a common climate signal globally using deviations from the average climate at the mortality plot locations using a gridded ~4 km² monthly climate dataset extrapolated from local weather stations (TerraClimate), and to project how much more common the climate signal would become under future warming using simple adjustments of the past gridded ~4 km² monthly climate dataset. The study found that there was a common climate signal that corresponded with mortality events using all of the > 600 cells that corresponded with the gridded ~4 km² monthly climate dataset. That common signal was that all six of the indicators used (monthly average maximum temperature, vapor pressure deficit, climatic water deficit, soil moisture, precipitation, and the Palmer Drought Severity Index) differed significantly from their long-term means, for all data combined. The finding that the year before and after the mortality initiation had climate similar to the year of mortality initiation is interesting. However, the common signal did not work well for the Whittaker Biomes with fewer plots (tropical seasonal forest/savanna, 8% of plots; temperate grassland/desert, 12% of plots; Tropical rainforest and boreal forest, together < 4% of plots); these all had one or more of the six indicators not different from the long-term average. The simplified warming estimation increased the frequency of the hot-drought indicators by 22% for the +2°C scenario and by 140% for the +4°C scenario.

This is an important and impactful paper. Assembling the published tree mortality data for public access is a large, novel advance, as is its use in linking it with the TerraClimate monthly climate data to begin to identify climate correlates of drought-related tree mortality. The finding that all six of the climate correlates needed to be beyond their 1958-present mean values to indicate a large mortality event is striking and important.

I have some suggestions that would make the study and manuscript more impactful.

1. The analysis to determine the climate "fingerprint" of large tree mortality events is perhaps too simplistic. Of course, it was limited by the information available in the TerraClimate database and much of the information available was used. But, there was no description of how the specific variables used were selected from a larger suite of variables, and no analysis of how the six indicators estimated large mortality events for the individual sites. How many of the mortality events at the >600 sites were matched with all six indicators? How many sites with mortality events did not have all or most of the six indicator variables? Presenting the statistics by site would be as or more informative than the global means and CIs. Figure S6 should be included in the main text – it is a more realistic of how the climate indicators work and where they work.
2. There was no quantitative comparison of this study's results with those from projections identified in the Introduction and Discussion. Such comparison would be extremely useful.
3. I understand why the warming scenarios were done as they were, but some comparison of these scenarios with GCM estimates would help convince the reader that they are reasonable to do.
4. The text could be shortened considerably, particularly the more speculative sections in the Discussion. There is also much repetition that could be removed. The Results section lacks enough detail about how the analysis was done to make the Results a stand alone section.
5. Figures 4 and 5. The TerraClimate website specifically states that the TerraClimate data is specifically NOT supposed to be used for these sorts of analyses, because of its limitations.
6. Beyond stating that the mortality event exceeded 15% of stems, there is no information on

magnitude of mortality or prevalence of impact to different size trees. Mortality of 15% of the larger stems would have a much larger impact than 15% mortality of smaller trees.

L43-44. The "up 140% by +4°C" is impossible to understand without reading the paper first.

L157-160. 'Monthly average' does not apply to precipitation. Also, Is the VPD 24 hour or only in the light period?

L238-241. This sentence is very difficult to understand.

Reviewer #2:

Remarks to the Author:

I really like this study and I acknowledge the importance. I have some general concerns about global applicability given the biases of the study sites:

1. Most peer-reviewed research sites with this type of data are limited to Northern Hemisphere/well-funded places.
2. "By Whittaker biome woodland/shrubland accounted for 49% of plots (n=638) which spans a large climatic niche". I agree that some of these might be 'forests' but it really does reduce the sample size (by half) for the forests/trees that are of most concern for climate change (the forest that are actually carbon sinks).
3. "Tropical rainforest, temperate rainforest, and boreal forest biomes combined contained just 4% of plots". As these are the most non-drought adapted forests, this seems like a huge data deficit!

Those concerns aside, the study does characterize what is actually available. The database is a critical step forward.

My second concern is that the database will 'help improve models'. Because there are no mechanisms associated with the study, the database can really only be used to validate model projections given certain climate conditions (those met in the study for mortality). I suggest the authors consider/learn how tree mortality is actually performed in ESMs and then consider how this database is useful.

Finally, given the limitations, it seems that one of the largest contributions this study has is to once again point out that we need more studies funded to document and study tree mortality, especially in forests important for carbon sequestration and may be more sensitive to drought. This is alluded to in the final sections but could be pointed out more.

Response to Reviewers

“Global field observations of tree die-off reveal hotter-drought fingerprint for Earth’s forests” Hammond *et al.*

REVIEWER COMMENTS

Reviewer #1:

Comment: Reviewer #1 (Remarks to the Author):

The aim of this study and manuscript was to describe a database of tree mortality studies in published literature, to assess if there was a common climate signal globally using deviations from the average climate at the mortality plot locations using a gridded ~4 km² monthly climate dataset extrapolated from local weather stations (TerraClimate), and to project how much more common the climate signal would become under future warming using simple adjustments of the past gridded ~4 km² monthly climate dataset. The study found that there was a common climate signal that corresponded with mortality events using all of the > 600 cells that corresponded with the gridded ~4 km² monthly climate dataset. That common signal was that all six of the indicators used (monthly average maximum temperature, vapor pressure deficit, climatic water deficit, soil moisture, precipitation, and the Palmer Drought Severity Index) differed significantly from their long-term means, for all data combined. The finding that the year before and after the mortality initiation had climate similar to the year of mortality initiation is interesting. However, the common signal did not work well for the Whittaker Biomes with fewer plots (tropical seasonal forest/savanna, 8% of plots; temperate grassland/desert, 12% of plots; Tropical rainforest and boreal forest, together < 4% of plots); these all had one or more of the six indicators not different from the long-term average. The simplified warming estimation increased the frequency of the hot-drought indicators by 22% for the +2 °C scenario and by 140% for the +4 °C scenario.

This is an important and impactful paper. Assembling the published tree mortality data for public access is a large, novel advance, as is its use in linking it with the TerraClimate monthly climate data to begin to identify climate correlates of drought-related tree mortality. The finding that all six of the climate correlates needed to be beyond their 1958-present mean values to indicate a large mortality event is striking and important.

I have some suggestions that would make the study and manuscript more impactful.

Response:

Thank you for your summary of our findings and assessment that ours is an important and impactful paper, and your constructive comments and suggestions. Below, we provide point-specific responses to your comments and suggestions.

Comment :

1. The analysis to determine the climate "fingerprint" of large tree mortality events is perhaps too simplistic. Of course, it was limited by the information available in the TerraClimate database and much of the information available was used. But, there was no description of how the specific variables used were selected from a larger suite of variables, and no analysis of how the six indicators estimated large mortality events for the individual sites. How many of the mortality events at the >600 sites were matched with all six indicators? How many sites with mortality events did not have all or most of the six indicator variables? Presenting the statistics by site

would be as or more informative than the global means and CIs. Figure S6 should be included in the main text – it is a more realistic of how the climate indicators work and where they work.

Response:

While a more complex approach can always be taken, our direct empirical approach was to focus on climatically-important variables already widely recognized as related to hotter-drought induced tree mortality. To address your specific concerns regarding variable selection and site-level statistics, we have:

1. Included a statement on variable selection which is now included in the results section of the main text:
“We chose these six climate variables *a priori* from the TerraClimate database due to their being direct or indirect measures for heat and/or drought impacts.” now on lines (158-159). Thus, we did not include other variables (e.g., minimum temperature, wind speeds, solar radiation) from TerraClimate not relevant to hotter-drought, and thank the reviewer for pointing out the value of explaining our selection of variables.
2. We also now provide a summary of the number of sites that had 1, 2, 3, 4, 5, or all 6 of the climatic variables in common ('hotter' and/or 'drier' than long-term means, as defined in the main text) during the mortality year textually: “At the site level, conditions in the mortality year were anomalously hot/dry for all six variables at 24% of sites, for at least 5 variables at 47% of sites, and for at least 4 variables at 69% of sites (Fig. S2).” (now on lines 195-197) as suggested, and further provide details as supplemental figure S2 which is shown in this response letter below (along with its caption, for convenience), and can be found now on lines (777-786) of the supplemental figures document.

(NEW) “Figure S2. Number of sites with climate variables exceeding local long-term trends (e.g., combined ‘hotter-drier’ climate variables).

The number of sites that had 0, 1, 2, 3, 4, 5, or all 6 of the climatic variables in common ('hotter' and/or 'drier' than long-term means, as defined in the main text) during the mortality year. At the site level, 164 sites (24%) exceeded their long-term (1958-2019) climate average for all six variables (hotter AND drier) during the mortality year, while 317 sites (47%) had at least 5 concurrent variables hotter/drier, and 467 sites (69%) had 4 or more climate variables exceeding long-term means.” (Added on lines 777-786)

3. We agree that Figure S6 (the by-biome fingerprint figure) represents the nuance of where the fingerprint works best, and where it does not (along with highlighting the available per-biome sample sizes of data, and especially biomes where more sampling is needed, as suggested by reviewer 2). Thus, we have moved this figure into the main text as a new **Figure 4** per reviewer 1's suggestion (now on line 724-733).

Comment :

2. *There was no quantitative comparison of this study's results with those from projections identified in the Introduction and Discussion. Such comparison would be extremely useful.*

Response:

In addition to our prior statement comparing our study's results with those from projections we earlier referenced, for example: "While our projections for an increase by up to 140% in the frequency of climate conditions associated with recent forest die-off under +4 °C may seem severe, they are modest in comparison to some current empirical and mechanistic process-based model predictions for catastrophic forest die-off at continental scales under hotter droughts" on lines 300-302 of the manuscript, we have also now added to the discussion that:

"Our projections for increasing die-offs under further warming are consistent with projections showing the potential for large increases in mortality under future hotter-drought (Adams *et al.*, 2009, 2017; McDowell *et al.*, 2016), although these projections are often limited to single-species or single biomes. Even continental-scale projections for up to 40% increases in the frequency of mortality-inducing hotter-droughts under $\sim +2.5\text{C}$ since pre-industrial (Mitchell *et al.*, 2014) are in general agreement with our global analysis's 20% under +2C (Figure 6d)." (Now on lines 303-309).

While presently there still are no globally-scaled process-model projections of tree mortality, global projections of carbon turnover are in high disagreement — likely reflecting large but undetermined underlying between-model differences in vegetation (tree) mortality — with uncertainty equal to fifty years of anthropogenic emissions (Sitch *et al.*, 2008; Pugh *et al.*, 2020). Current model projections of tree mortality are generally so single-taxa derived or single-biome focused (McDowell *et al.*, 2016; Adams *et al.*, 2017; Law *et al.*, 2019) that a direct quantitative comparison with our present global study's results is not possible. Other efforts (e.g., (Jiang *et al.*, 2013) have used bioclimatic limits to forest functional types to make projections of forest conversions to other vegetation, but these approaches are not based on the climate conditions that specifically cause tree die-off. Although note the example listed above, and now in-text, of continental-scale projections of Mitchell *et al.*, 2014. which represents the as-of-yet largest-scale attempt at identifying climatic thresholds, across the Australian continent. To our knowledge, no other study has made global projections of tree mortality based on the species-specific conditions that kill trees from a variety of species. Making our globally-extensive and species-rich (n=157 species) dataset available will enable such quantitative comparisons, but the studies we have referenced lacked the diversity in species, space, and time present in our database. Thus, we have caveated our prior mentions of these studies projections to make it clear that they were not similarly globally extensive or species-rich as in the present study (see new text, above). Nonetheless, our results are consistent with single-taxa, single-biome, and plant functional-type projections for massive increases in mortality under further warming.

Comment:

3. I understand why the warming scenarios were done as they were, but some comparison of these scenarios with GCM estimates would help convince the reader that they are reasonable to do.

Response:

We agree that comparing these warming scenarios with recent GCM estimates strengthens our argument, and so have added a comparison textually (now on lines 471-474) as suggested by the reviewer. The added text now states: "These levels of warming are consistent with GCM ensemble estimates for "middle of the road" (e.g., SSP2-4.5) which could result in > +2C warming by 2100, and "fossil fueled development" (SSP5-8.5) which could result in > +4C warming by 2100 (O'Neill *et al.*, 2016)."

Comment:

4. The text could be shortened considerably, particularly the more speculative sections in the Discussion. There is also much repetition that could be removed. The Results section lacks enough detail about how the analysis was done to make the Results a stand alone section.

Response:

We have removed repetition from many places in the manuscript's discussion (for example, repeating the integrative nature of our empirical approach on lines (e) of the prior submission was removed, and 9 lines of discussion that were more speculative regarding the influence of CO₂ on tree mortality were also removed). In total, we removed ~19 lines of the discussion while trimming repetitive and/or speculative portions, as suggested by the reviewer. This provided room also to add details to the results section, where we added ~18 lines as suggested by the reviewer. In total, our revised text is only 27 words longer than the originally submitted manuscript, but this significant re-balancing (to reduce discussion repetitions, and to increase details in the results section) allows the results to be a stand alone section, as suggested by the reviewer.

Comment:

5. Figures 4 and 5. The TerraClimate website specifically states that the TerraClimate data is specifically NOT supposed to be used for these sorts of analyses, because of its limitations.

Response:

We appreciate the reason for this comment from the reviewer and agree that data limitations should always be considered, and analyses conducted within data limitations. However, please note that the author of the guidelines on the TerraClimate website that is referenced in this reviewer 1 concern is Dr. John Abatzoglou, a key coauthor of this study who centrally participated in the data analysis and interpretations. In discussing this reviewer comment with Dr. Abatzoglou, he emphasizes that specifically, the TerraClimate website states that: "Long-term trends in data such as temperature and precipitation are inherited from parent datasets. TerraClimate should not be used directly for **independent** assessments of trends relative to these parent datasets." Here, we are not comparing TerraClimate trends to those from other CRU-derived datasets, as warned against in the data limitations statement. This data limitations statement reflects the fact that TerraClimate was not created using data independent from other

sources and hence should not be treated as such in multi-dataset assessment of trends (e.g., comparison of TerraClimate vs. CRU, as TerraClimate uses CRU). Nowhere is it stated that these data are not intended for the sorts of analyses in the current paper, and John specifically confirms that our use of this dataset that he led the development of (Abatzoglou et al. 2018) is appropriate. Indeed, numerous other high-impact studies have appropriately used TerraClimate in analyses similar to that conducted herein (Zellweger *et al.*, 2020; Cook-Patton *et al.*, 2020), including publication in *Nature Communications* (Berner *et al.*, 2020). So we affirm that the use of TerraClimate data to develop Figures 4 and 5 was methodologically valid.

In a related further follow-up to reviewer #1's first comment (above), requesting more site-level information about the fingerprint, we have decided to keep these figures as they are the only figures in the paper which show data for each of the 675 sites. Figure 5 will be kept in the main text, while Figure 4 is being moved to the supplement (to make room for the inclusion of the new per-biome figure, another request of the reviewer). Additionally, we have added a new supplemental figure that describes the number of climate variables that met 'hotter/drier' conditions at the site-level. Given our justification for not removing these figures (see above response about the guidance on TerraClimate website), we do believe that retention of these figures (Fig. 5 in the main text, Fig. 4 in the supplemental) is also helpful in demonstrating the "per site" performance of our indicator metrics—as each point represents the anomaly of an individual site in our global analysis. We now reference this supplemental figure in the main text after the added points to address your prior comment, regarding the desire to see a summary of more site-specific results, which has been added to the results section of our paper to state:

“At the site level, conditions in the mortality year were anomalously hot/dry for all six variables at 24% of sites, for at least 5 variables at 47% of sites, and for at least 4 variables at 69% of sites (Fig. S2).” (Added on lines 195-197)

We decided to keep figure 5 (Mortality year TMAX is warming faster than all-years) in the main text, as we believe that the increase in mortality-year TMAX anomalies is helpful in highlighting why temperature extremes should be further considered in studies of climate-induced mortality, which to date have had a strong bias for the effects of drought, relative to temperature. To further emphasize this, we have added additional text in the results section:

At our sites, TMAX has been increasing during mortality years faster than background warming (Fig. 5), consistent with continental-scale observations of intensifying extremes (Alizadeh *et al.*, 2020). (Added on lines 204-206).

We hope that you will agree with its retention in the manuscript.

Comment:

6. Beyond stating that the mortality event exceeded 15% of stems, there is no information on magnitude of mortality or prevalence of impact to different size trees. Mortality of 15% of the larger stems would have a much larger impact than 15% mortality of smaller trees.

Response:

This is an important point, and we have added additional detail to our database section, to make it clear that 15% standing dead was for "mature trees". We also acknowledge some variability in

precise definitions between studies, even regarding what constitutes a ‘tree’ and not a ‘sapling’ or ‘seedling’. This sentence now reads: “

When mortality was reported as a percentage, we determined ‘higher-than-expected’ mortality based on authors’ expertise of their study system. Often, ‘standing dead’ of 15% or greater (for mature trees) was used, as 2-3% annual background mortality rates accumulated for up to five years would not be expected to exceed this threshold

(Now on lines 443-447).

In addition to our section detailing the database, we know in some but not all cases that larger trees are most negatively affected (Nepstad *et al.*, 2007; Rowland *et al.*, 2015; Bennett *et al.*, 2015; Patrut *et al.*, 2018; Trugman *et al.*, 2018; Stovall *et al.*, 2019; Stephenson & Das, 2020). Size-class specific analyses were not possible given the many disparate sources and methods in collection of the data within our database, but generally the evidence supports our claim in the discussion of historical forests that during hotter-drought extremes :

...”large old trees—exemplars of Earth’s historical forests(Lutz *et al.*, 2018)—can be especially susceptible to mortality (Mcdowell & Allen, 2015; Rowland *et al.*, 2015; Bennett *et al.*, 2015; Lindenmayer & Laurance, 2017; Stephenson *et al.*, 2019)” (Now on lines 283-284).

Comment:

L43-44. The “up 140% by +4°C” is impossible to understand without reading the paper first.

Response:

We thank the reviewer for this comment. We have removed this phrase from the abstract, instead emphasizing that the increase is nonlinear with warming. The sentence now states:

“Frequency of these observed mortality-year climate conditions strongly increases nonlinearly under projected warming.” (Now on lines 40-41)

Comment:

L157-160. ‘Monthly average’ does not apply to precipitation. Also, Is the VPD 24 hour or only in the light period?

Response:

We thank the reviewer for catching this, we have added ‘monthly total’ before precipitation as suggested by the reviewer (now on line 156), and further clarify that VPD is a 24-hour metric (mean daily VPD) in the methods section of the paper, (now on line 460).

Comment:

L238-241. This sentence is very difficult to understand.

Response:

We have nearly completely rewritten this previously long and complicated sentence for clarity, and indeed thank the reviewer for pointing out that it was difficult to understand. It now reads:

“Other approaches could consider other temporal dimensions of climate signals (e.g., shorter-term heat-wave stress, longer-term chronic drought, changes in seasonal drought duration or timing), which may further improve our understanding of climatic drivers of tree mortality.”
Now on lines (239-241).

Reviewer #2:

Reviewer #2 (Remarks to the Author):

Comment:

I really like this study and I acknowledge the importance. I have some general concerns about global applicability given the biases of the study sites:

1. Most peer-reviewed research sites with this type of data are limited to Northern Hemisphere/well-funded places.

Response:

The proportion of sites in the northern hemisphere for our analysis is 0.70 (471 of the 675 plots). This information is now included in our manuscript along with a mention of the proportion of global forest cover (0.78) that is in the northern hemisphere. We calculated this as the proportion of all ‘forest’ pixels in the Simard *et al.* canopy height dataset (where forest is identified as canopy height ≥ 5 m, the green in our paper’s Figure 1) that have positive latitude. This is now in the manuscript in the results section, and states:

“While our database sites have a strong northern hemisphere bias (70%, $n=471$ of the 675 total sites), the total forested area in the northern hemisphere is $\sim 78\%$ of the global total (calculated from Simard *et al.*’s canopy height shown in Fig. 1). Even so, critically important carbon sinks, particularly boreal forests and both tropical and temperate rainforests, remain notably under-sampled.” (now on lines 130-134).

While there are many ways to define forests and quantify them, we nevertheless agree with the reviewer that mentioning this bias in our dataset is important. We also have expanded our discussion on future sampling needs to include awareness of northern hemisphere and/or well-funded research institution adjacency biases. It now states:

“Ideally, future efforts to harmonize global forest inventory and monitoring methodologies, including their currently-disparate documentation of tree mortality, will reduce the inherent sampling biases (typically favoring northern hemisphere and/or areas adjacent to well-funded research institutions) and presence-only limitations of our present database (Hartmann *et al.*, 2018).” (now on lines 242-245).

Comment:

2. *"By Whittaker biome woodland/shrubland accounted for 49% of plots (n=638) which spans a large climatic niche". I agree that some of these might be 'forests' but it really does reduce the sample size (by half) for the forests/trees that are of most concern for climate change (the forest that are actually carbon sinks).*

Response:

While our database's woodland/shrubland biome sample is large, we would like to refer the reviewer to the new to main-text Figure 4, a per-biome analysis (previously Supplemental Figure 6) which shows the three-year window, centered on the mortality start year, displayed for each biome. Above each biome triptych the number of sites included is listed. Here, you can see that significant hotter-drought signals occur not just in the woodland/shrubland, where our database is richest in observation, but similarly strong signals also appear in temperate and tropical seasonal forests, and even boreal forests (with a small sample)—all show significant indicators of hotter drought. We believe that by moving this figure into the main text, it provides the critical context that you rightly identified as missing—that the current best-documented "global signal" has a bias for the Whittaker-defined woodland/shrubland biome. But this figure also demonstrates the extensibility of this approach to additional biomes of various sampling. Also, we have added to the results that:

"Even so, critically important carbon sinks, particularly boreal forests and both tropical and temperate rainforests, remain notably under-sampled." (Now on lines 132-134).

Further, on lines (142-146), we have added detail to our existing statement on the complex composition of this 'woodland/shrubland' biome in the Whittaker scheme:

"We note that these coarse climate-based Whittaker biomes can obscure heterogeneous forest types within single biomes; in particular, the woodland/shrubland Whittaker biome is dominated in our database by diverse, relatively dry but often closed-canopy forest types, including those composed of aspen and numerous conifer, oak, and eucalypt species." We also refer the reviewer also to Supplementary Table 1, which includes the biome designations of plots from the 154 peer-reviewed studies—this table documents that this 'woodland/shrubland' biome includes diverse forest types with substantial biomass/carbon stocks, ranging from widespread dense montane pine and mixed-conifer forests in the western US, closed-canopy Oak/Hickory forests in the midwestern Ozarks of North America, pine-oak and Spanish fir in the southern Mediterranean, and closed-canopy Jarrah forests of SW Australia.

Comment:

3. *"Tropical rainforest, temperate rainforest, and boreal forest biomes combined contained just 4% of plots". As these are the most non-drought adapted forests, this seems like a huge data deficit! Those concerns aside, the study does characterize what is actually available. The database is a critical step forward.*

Response:

We agree. We have added the following text to the results following the presentation of the fingerprint:

"Even so, critically important carbon sinks, particularly boreal forests and both tropical and temperate rainforests, remain notably under-sampled." Now on lines (132-134).

Additionally, and in line with your final suggestion, we have added a new point to our “Roadmap for Research” to highlight the need to fill these critical data gaps. The additional text reads:

“(7) Invest in monitoring, documenting, and gathering mortality data for forests under-represented in this initial global database—especially in the extensive critical carbon sinks of boreal forests and tropical rainforests.” Now on lines (402-404).

Comment:

My second concern is that the database will 'help improve models'. Because there are no mechanisms associated with the study, the database can really only be used to validate model projections given certain climate conditions (those met in the study for mortality). I suggest the authors consider/learn how tree mortality is actually performed in ESMs and then consider how this database is useful.

Response:

We thank the reviewer for this comment, and have changed our prior claim of “The widespread global coherence of our empirically quantified hotter-drought fingerprint may provide immediate improvements for representing tree mortality in models of the Earth System...” to no longer claim that our database or emergent fingerprint analyses will directly improve such process models. Rather, we now state that the fingerprint:
...“may provide immediate opportunities to validate projections of tree mortality in existing models of the Earth system” ... as suggested by reviewer 2. This change is now on lines (340-342).

Comment:

Finally, given the limitations, it seems that one of the largest contributions this study has is to once again point out that we need more studies funded to document and study tree mortality, especially in forests important for carbon sequestration and may be more sensitive do drought. This is alluded to in the final sections but could be pointed out more.

Response:

Thank you for pointing out this useful of aspect of our study — that the spatial variability of observations in our Figure 1 can provide a roadmap for future data collection, highlighting undersampled areas that need many more observations. To that end, we have added a 7th point to our “Roadmap for research” that states:
“(7) Invest in monitoring, documenting, and gathering mortality data for forests under-represented in this initial global database—especially in the extensive critical carbon sinks of boreal forests and tropical rainforests.” Now on lines (402-404).

References for Response to Reviewers:

Adams HD, Barron-Gafford GA, Minor RL, Gardea AA, Bentley LP, Law DJ, Breshears DD, McDowell NG, Huxman TE. 2017. Temperature response surfaces for mortality risk of tree species with future drought. *Environmental Research Letters* **12**: 115014. DOI:10.1088/1748-9326/aa93be.

- Adams HD, Guardiola-Claramonte M, Barron-Gafford GA, Villegas JC, Breshears DD, Zou CB, Troch PA, Huxman TE. 2009.** Temperature sensitivity of drought-induced tree mortality portends increased regional die-off under global-change-type drought. *Proceedings of the National Academy of Sciences* **106**: 7063–7066. DOI:10.1073/pnas.0901438106.
- Alizadeh MR, Adamowski J, Nikoo MR, AghaKouchak A, Dennison P, Sadegh M. 2020.** A century of observations reveals increasing likelihood of continental-scale compound dry-hot extremes. *Science Advances* **6**: eaaz4571. DOI:10.1126/sciadv.aaz4571.
- Bennett AC, McDowell NG, Allen CD, Anderson-Teixeira KJ. 2015.** Larger trees suffer most during drought in forests worldwide. *Nature Plants* **1**: 15139. DOI:10.1038/nplants.2015.139.
- Berner LT, Massey R, Jantz P, Forbes BC, Macias-Fauria M, Myers-Smith I, Kumpula T, Gauthier G, Andreu-Hayles L, Gaglioti BV, et al. 2020.** Summer warming explains widespread but not uniform greening in the Arctic tundra biome. *Nature Communications* **11**: 4621. DOI:10.1038/s41467-020-18479-5.
- Cook-Patton SC, Leavitt SM, Gibbs D, Harris NL, Lister K, Anderson-Teixeira KJ, Briggs RD, Chazdon RL, Crowther TW, Ellis PW, et al. 2020.** Mapping carbon accumulation potential from global natural forest regrowth. *Nature* **585**: 545–550. DOI:10.1038/s41586-020-2686-x.
- Hartmann H, Moura CF, Anderegg WR, Ruehr NK, Salmon Y, Allen CD, Arndt SK, Breshears DD, Davi H, Galbraith D. 2018.** Research frontiers for improving our understanding of drought-induced tree and forest mortality. *New Phytologist* **218**: 15–28.
- Jiang X, Rauscher SA, Ringler TD, Lawrence DM, Williams AP, Allen CD, Steiner AL, Cai DM, McDowell NG. 2013.** Projected Future Changes in Vegetation in Western North America in the Twenty-First Century. *Journal of Climate* **26**: 3671–3687. DOI:10.1175/JCLI-D-12-00430.1.
- Law DJ, Adams HD, Breshears DD, Cobb NS, Bradford JB, Zou CB, Field JP, Gardea AA, Williams AP, Huxman TE. 2019.** Bioclimatic Envelopes for Individual Demographic Events Driven by Extremes: Plant Mortality from Drought and Warming. *International Journal of Plant Sciences* **180**: 53–62. DOI:10.1086/700702.
- Lindenmayer DB, Laurance WF. 2017.** The ecology, distribution, conservation and management of large old trees: Ecology and management of large old trees. *Biological Reviews* **92**: 1434–1458. DOI:10.1111/brv.12290.
- Lutz JA, Furniss TJ, Johnson DJ, Davies SJ, Allen D, Alonso A, Anderson-Teixeira KJ, Andrade A, Baltzer J, Becker KML, et al. 2018.** Global importance of large-diameter trees. *Global Ecology and Biogeography* **27**: 849–864. DOI:10.1111/geb.12747.
- McDowell NG, Allen CD. 2015.** Darcy's law predicts widespread forest mortality under climate warming. *Nature Climate Change* **5**: 669–672. DOI:10.1038/nclimate2641.
- McDowell NG, Williams AP, Xu C, Pockman WT, Dickman LT, Sevanto S, Pangle R, Limousin J, Plaut J, Mackay DS. 2016.** Multi-scale predictions of massive conifer mortality due to chronic temperature rise. *Nature Climate Change* **6**: 295.
- Mitchell PJ, O'Grady AP, Hayes KR, Pinkard EA. 2014.** Exposure of trees to drought-induced die-off is defined by a common climatic threshold across different vegetation types. *Ecology and Evolution* **4**: 1088–1101. DOI:10.1002/ece3.1008.
- Nepstad DC, Tohver IM, Ray D, Moutinho P, Cardinot G. 2007.** MORTALITY OF LARGE TREES AND LIANAS FOLLOWING EXPERIMENTAL DROUGHT IN AN AMAZON FOREST. *Ecology* **88**: 2259–2269. DOI:10.1890/06-1046.1.

- O'Neill BC, Tebaldi C, van Vuuren DP, Eyring V, Friedlingstein P, Hurtt G, Knutti R, Kriegler E, Lamarque J-F, Lowe J, et al. 2016.** The Scenario Model Intercomparison Project (ScenarioMIP) for CMIP6. *Geoscientific Model Development* **9**: 3461–3482. DOI:10.5194/gmd-9-3461-2016.
- Patrut A, Woodborne S, Patrut RT, Rakosy L, Lowy DA, Hall G, von Reden KF. 2018.** The demise of the largest and oldest African baobabs. *Nature Plants* **4**: 423–426. DOI:10.1038/s41477-018-0170-5.
- Pugh TAM, Rademacher T, Shafer SL, Steinkamp J, Barichivich J, Beckage B, Haverd V, Harper A, Heinke J, Nishina K, et al. 2020.** Understanding the uncertainty in global forest carbon turnover. *Biogeosciences* **17**: 3961–3989. DOI:10.5194/bg-17-3961-2020.
- Rowland L, Da Costa ACL, Galbraith DR, Oliveira RS, Binks OJ, Oliveira AAR, Pullen AM, Doughty CE, Metcalfe DB, Vasconcelos SS, et al. 2015.** Death from drought in tropical forests is triggered by hydraulics not carbon starvation. *Nature* **528**: 119–122. DOI:10.1038/nature15539.
- Sitch S, Huntingford C, Gedney N, Levy PE, Lomas M, Piao SL, Betts R, Ciais P, Cox P, Friedlingstein P, et al. 2008.** Evaluation of the terrestrial carbon cycle, future plant geography and climate-carbon cycle feedbacks using five Dynamic Global Vegetation Models (DGVMs). *Global Change Biology* **14**: 2015–2039. DOI:10.1111/j.1365-2486.2008.01626.x.
- Stephenson NL, Das AJ. 2020.** Height-related changes in forest composition explain increasing tree mortality with height during an extreme drought. *Nature Communications* **11**. DOI:10.1038/s41467-020-17213-5.
- Stephenson NL, Das AJ, Ampersee NJ, Bulaon BM, Yee JL. 2019.** Which trees die during drought? The key role of insect host-tree selection (D Edwards, Ed.). *Journal of Ecology* **107**: 2383–2401. DOI:10.1111/1365-2745.13176.
- Stovall AEL, Shugart H, Yang X. 2019.** Tree height explains mortality risk during an intense drought. *Nature Communications* **10**. DOI:10.1038/s41467-019-12380-6.
- Trugman AT, Detto M, Bartlett MK, Medvigy D, Anderegg WRL, Schwalm C, Schaffer B, Pacala SW. 2018.** Tree carbon allocation explains forest drought-kill and recovery patterns (D Cameron, Ed.). *Ecology Letters* **21**: 1552–1560. DOI:10.1111/ele.13136.
- Zellweger F, De Frenne P, Lenoir J, Vangansbeke P, Verheyen K, Bernhardt-Römermann M, Baeten L, Hédli R, Berki I, Brunet J, et al. 2020.** Forest microclimate dynamics drive plant responses to warming. *Science* **368**: 772–775. DOI:10.1126/science.aba6880.

Reviewers' Comments:

Reviewer #1:

Remarks to the Author:

This manuscript is a re-review of a submission that I reviewed in August, 2021. I have read the response to the reviewers and the revised manuscript, and the authors did an excellent job of responding to the reviewers' questions and suggestions. The authors have convincingly revised the manuscript in response to reviewer suggestions. I repeat my statement about the worth of this manuscript from my original review: "This is an important and impactful paper. Assembling the published tree mortality data for public access is a large, novel advance, as is its use in linking it with the TerraClimate monthly climate data to begin to identify climate correlates of drought-related tree mortality. The finding that all six of the climate correlates needed to be beyond their 1958-present mean values to indicate a large mortality event is striking and important."

REVIEWERS' COMMENTS

Reviewer #1 (Remarks to the Author):

This manuscript is a re-review of a submission that I reviewed in August, 2021. I have read the response to the reviewers and the revised manuscript, and the authors did an excellent job of responding to the reviewers' questions and suggestions. The authors have convincingly revised the manuscript in response to reviewer suggestions. I repeat my statement about the worth of this manuscript from my original review: "This is an important and impactful paper. Assembling the published tree mortality data for public access is a large, novel advance, as is its use in linking it with the TerraClimate monthly climate data to begin to identify climate correlates of drought-related tree mortality. The finding that all six of the climate correlates needed to be beyond their 1958-present mean values to indicate a large mortality event is striking and important."

Response:

We appreciate the reviewer's time and attention in providing us the opportunity to strengthen our manuscript prior to publication. As the reviewer stated "The authors have convincingly revised the manuscript in response to reviewer suggestions", we have made no further changes according to the reviewer's comment here. However, we have made minor revisions per the requests of the editorial staff, and those are detailed in the attached "Author checklist" where there are point-by-point responses to the editorial office's requests.